Method

# Genome-wide profiling of histone modifications in *Plasmodium falciparum* using CUT&RUN

Riward Campelo Morillo[1] , Chantal T Harris[1,2], Kit Kennedy[1], Samuel R Henning[1], Björn FC Kafsack[1]

**We recently adapted a CUT&RUN protocol for genome-wide profiling of chromatin modifications in the human malaria parasite *Plasmodium*. Using the step-by-step protocol described below, we were able to generate high-quality profiles of multiple histone modifications using only a small fraction of the cells required for ChIP-seq. Using antibodies against two commonly profiled histone modifications, H3K4me3 and H3K9me3, we show here that CUT&RUN profiling is highly reproducible and closely recapitulates previously published ChIP-seq-based abundance profiles of histone marks. Finally, we show that CUT&RUN requires substantially lower sequencing coverage for accurate profiling compared with ChIP-seq.**

## Introduction

Post-translational histone modifications play a central role in regulating chromatin accessibility and partitioning in nearly all eukaryotes. Numerous modifications have been described with a wide variety of functions and dynamics reviewed in Rothbart and Strahl [2014]. The ability to profile these modifications genome-wide is a key tool for assigning their role in regulating gene expression and, more generally, for investigating nuclear functions. Traditionally, this has been accomplished by chromatin immuno-precipitation (ChIP) (Solomon et al, 1988), which relies on affinity-based pull-down from randomly fragmented chromatin of protein–DNA complexes with antibodies specific to one of the complex components followed by the extraction of the genomic fragments. High-throughput sequencing (HTS) methods are then used to determine the relative population abundance of the antibody target at each genomic position based on its relative enrichment in the precipitated chromatin versus its genomic frequency.

ChIP-seq has several drawbacks that derive from the random fragmentation of chromatin and the requirement for stringency during immunoprecipitation. The former results in relatively high background, although both make profiling low-affinity interactions difficult. This can be overcome by protein–DNA cross-linking before fragmentation. However, cross-linking presents another challenge as it can lead to non-specific recovery of untargeted proteins correlated with duration of fixation treatment (Baranello et al, 2016) and interfere with antibody recognition by epitope disruption (O'Neill & Turner, 2003).

The recent development of Cleavage Under Targets & Release Using Nuclease (CUT&RUN) addresses many of these challenges (Fig 1) (Skene et al, 2018; Meers et al, 2019). Instead of isolating and randomly fragmenting chromatin, cells are immobilized on beads and permeabilized to allow antibodies to diffuse into the nucleus to bind their targets in situ. After removal of excess antibodies, a fusion of protein A/G, which binds the Fc portion of antibodies, and micrococcal nuclease (MNase) is added under low calcium conditions that keep the MNase inactive. After the removal of excess protein A/G–MNase, calcium is added to activate MNase cleavage, thereby releasing the antibody-bound nucleosomes but leaving the unbound fraction of the genome intact. Because of their small size, liberated nucleosomes can then diffuse out of the cell, whereas the rest of the genome remains cell associated. DNA can then be extracted from the supernatant for standard HTS library generation, high-throughput sequencing, and analysis. Because of this, CUT&RUN has exceptionally low background and has been successfully used to profile chromatin interactions from very small number of cells (Liu et al, 2018; Skene et al, 2018). CUT&RUN has been successfully used for profiling a wide array of chromatin proteins in many experimental systems. In this study, we specifically focus on its use for profiling post-translational histone modifications on nucleosome.

We recently adapted a CUT&RUN protocol (Meers et al, 2019; Janssens & Henikoff, 2019 *Preprint*) for genome-wide profiling of chromatin modifications in the human malaria parasite *Plasmodium falciparum* (Harris et al, 2022 *Preprint*). Using the step-by-step protocol described below, we were able to generate high-quality profiles of multiple histone modifications using only a small fraction of the cells required for ChIP-seq. Using antibodies against two commonly profiled histone modifications, H3K4me3 and H3K9me3, we show here that CUT&RUN profiling is highly reproducible (Figs 2 and S1) and closely recapitulates previously published ChIP-seq-based abundance profiles of histone marks (Figs 3,

[1]Department of Microbiology and Immunology, Weill Cornell Medicine, New York, NY, USA    [2]Immunology and Microbial Pathogenesis Program, New York, NY, USA

Correspondence: rac2032@med.cornell.edu; bjk2007@med.cornell.edu

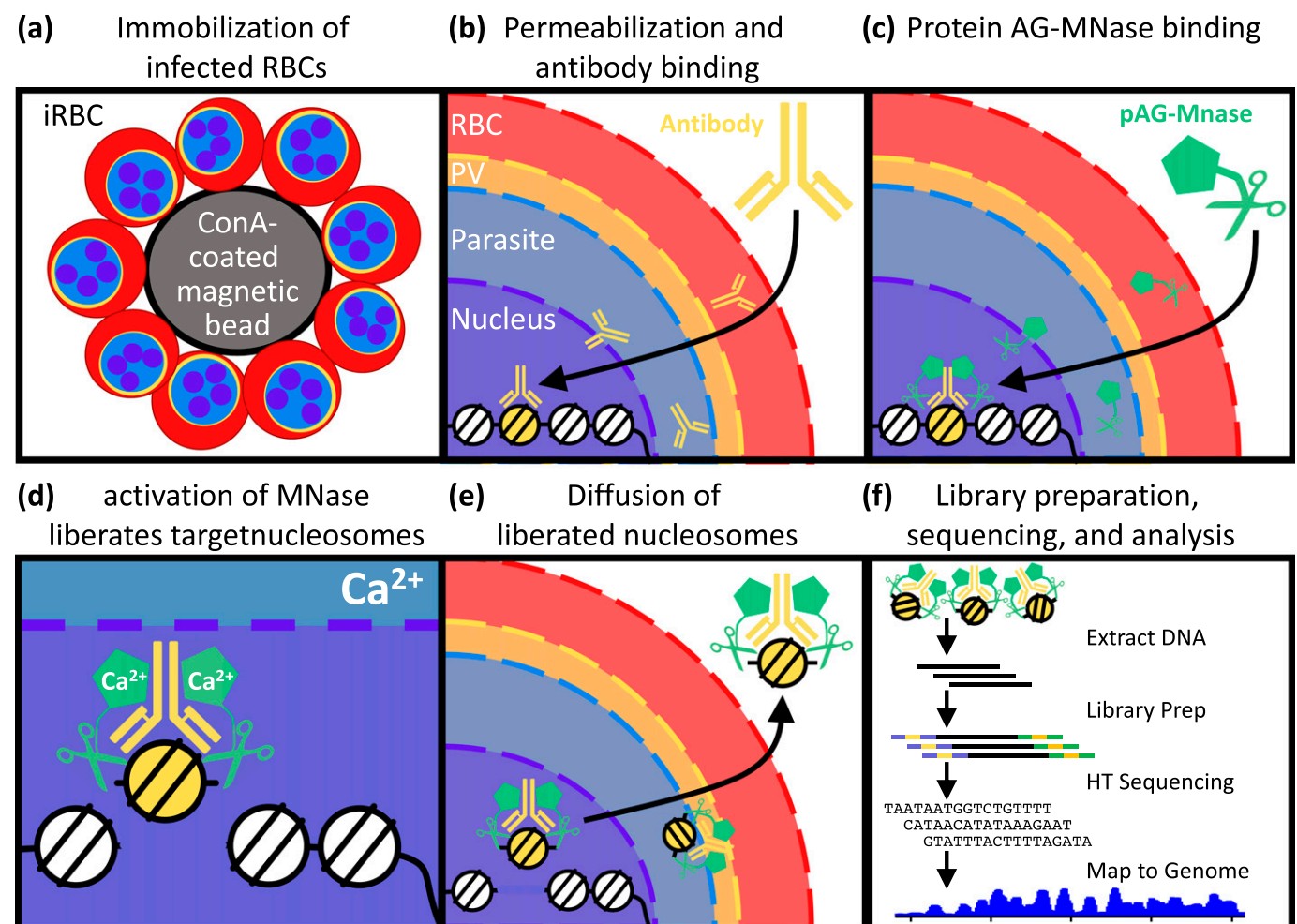

**Figure 1. Overview of genome-wide profiling of histone modifications using CUT&RUN.**
**(A)** Infected RBCs are immobilized on concanavalin-coated beads. **(B)** Immobilized cells are permeabilized, allowing histone modification-specific antibody to diffuse into the nucleus to bind to its targets. **(C)** After removal of excess antibodies, MNase fused to protein A/G is added under low calcium conditions and binds to chromatin-associated antibodies. **(D)** After removal of excess proteinAG–MNase, cleavage adjacent to bound nucleosomes is activated by addition of calcium-cations. **(E, F)** Liberated nucleosomes are allowed to diffuse out of the permeabilized cells for subsequent (F) DNA extraction, library generation, high-throughput sequencing, and analysis.

S1, and S2) (Bártfai et al, 2010; Bunnik et al, 2018; Fraschka et al, 2018). Finally, we show that CUT&RUN requires substantially lower sequencing coverage for accurate profiling compared with ChIP-seq (Fig 4).

## Planning and preparation

### *Prepare stock solutions*
**Timing: 4 h** 1 M HEPES-NaOH, pH 7.5: dissolve 238.3 g HEPES in 800 ml of distilled water. Adjust pH to 7.5 using 1 M NaOH, then adjust the final volume with distilled water to 1 liter. Store at RT.

1 M HEPES-KOH: dissolve 238.3 g HEPES in 800 ml of distilled water. Adjust pH to 7.5 using 1 M KOH, then adjust the final volume with distilled water to 1 liter. Store at RT.

1 M KCl: dissolve 74.55 g KCl in 800 ml of distilled water and adjust the volume to 1 liter. Store at RT.

5% sorbitol: dissolve 50 g sorbitol in 800 ml of distilled water and adjust the volume to 1 liter. Filter, sterilize, and store at RT.

90% percoll with 6% sorbitol: mix 20 ml of minimal RPMI with 180 ml percoll and dissolve 12 g D-sorbitol. Filter, sterilize, and store at 4°C.

100 mM $CaCl_2$: dissolve 1.1 g $CaCl_2$ in 100 ml of distilled water.

100 mM $MnCl_2$: dissolve 1.25 g $MnCl_2$ in 100 ml of distilled water.

100 mM EGTA, pH 8.0: dissolve 3.8 g EGTA in 80 ml of distilled water while maintaining the pH at 8.0 using 10 N NaOH. Adjust volume to 100 ml and store at RT.

500 mM EDTA, pH 8.0: dissolve 186.1 g EDTA in 800 ml of distilled water while maintaining the pH at 8.0 using 10 N NaOH. Adjust volume to 1 liter and store at RT.

1 M Tris–HCL, pH 8.0: dissolve 121.14 g Tris in 800 ml of distilled water. Adjust pH to 8.0 using concentrated HCl/10 N

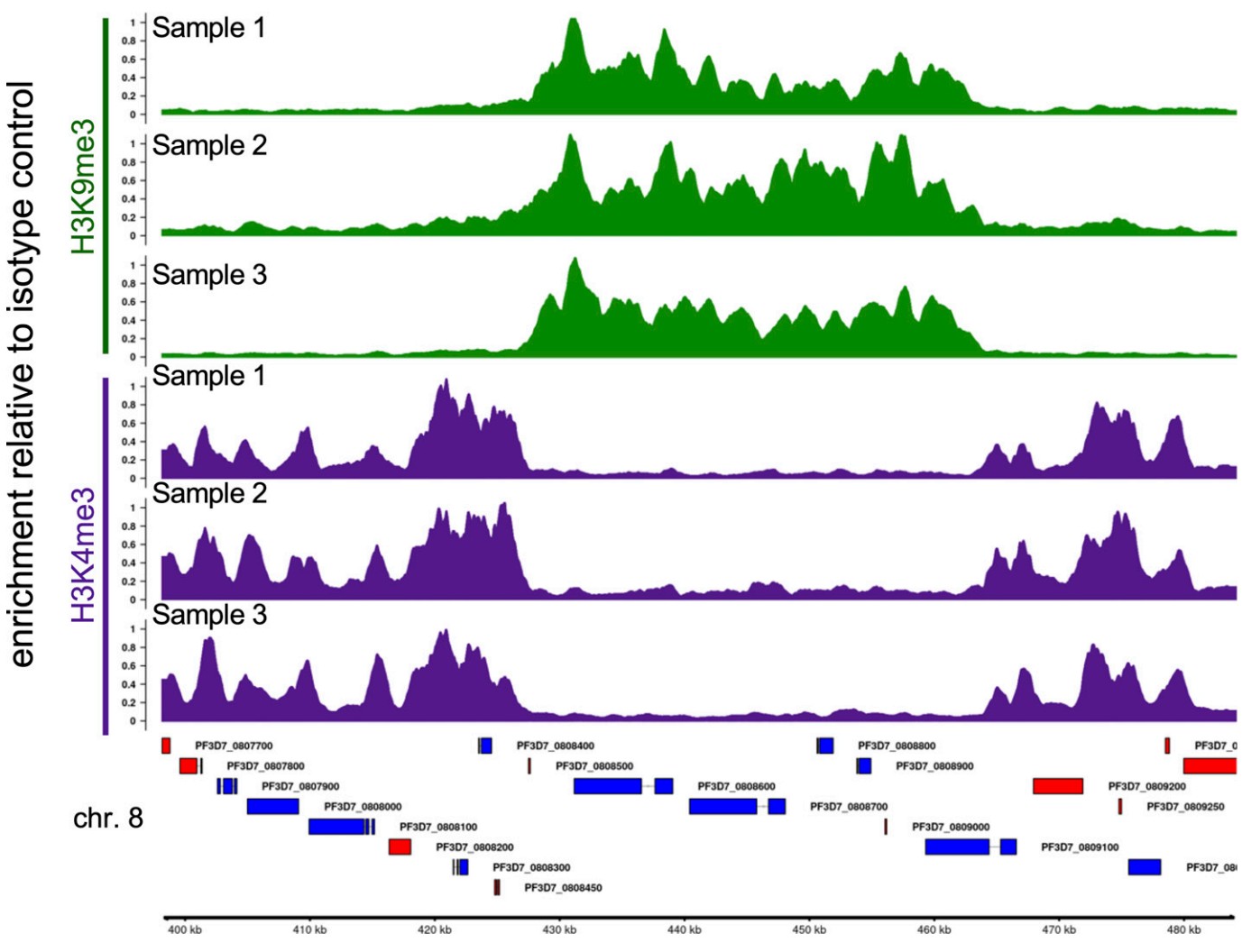

**Figure 2.  CUT&RUN profiles of histone modifications are highly reproducible.**
Fold Enrichment profiles of H3K9me3 (green) and H3K4me3 (purple) in three biological replicates. H3K4me3 and H3K9me3 tracks were scaled to maximal value in the region shown. Example locus on chromosome 8 contains a non-subtelomeric heterochromatin island. Genes shown in blue are encoded on the top (+) strand, whereas those in red are encoded on the bottom (−) strand. Gene IDs are shown to the right of the gene.

NaOH. Adjust the final volume with distilled water to 1 liter and store at RT.

5 M NaCl: dissolve 292.2 g NaCl in 800 ml distilled water. Adjust the volume to 1 liter and store at RT.

5% digitonin: dissolve 0.05 g of digitonin in 1 ml of DMSO. Prepare 150 μl aliquots and store at −20°C for up to 6 mo.

25× Protease Inhibitor Cocktail: dissolve one EDTA-free cOmplete protease inhibitor tablet in 1 ml of distilled water. Split into 200 μl aliquots and store at −20°C for up to 3 mo.

2 M spermidine: prepare a fresh work solution each time by diluting 31.25 μl of stock spermidine (6.4 M) in 100 μl of distilled water.

**P. falciparum *culturing and synchronization of erythrocytic stages***
**Timing: ~1 wk** We have found that CUT&RUN using parasites containing 10–50 million nuclei yielded robust signal for the antibodies tested. At a parasitemia and hematocrit of 5% each, this is

the equivalent of 0.4–2 ml of culture for 1 N parasites (ring, early trophozoite, and gametocyte stages) or as little as 20–100 μl of culture containing segmenting schizonts.

(1) Maintain parasite cultures in T25 flasks following established culturing techniques (Moll et al, 2013) using red blood cells at 3–5% hematocrit resuspended in standard malaria complete media.
(2) Double-synchronize parasite cultures with sorbitol treatment to achieve a synchrony of ± 6 h (Moll et al, 2013).

   (a) Wash away media by centrifugation at 800*g* × 3 min at RT.
   (b) Resuspend pelleted RBC by gently adding 10× volumes of 5% sorbitol.
   (c) Incubate for 5–10 min in a water bath at 37°C.
   (d) Spin down to wash away sorbitol and resuspend in fresh media.

**Reagents, materials, and equipment.**

| Reagent or resource | Source | Identifier |
|---|---|---|
| Antibodies | | |
| H3K9me3 rabbit polyclonal antibody ChIP grade | Abcam | ab8898 |
| H3K4me3 rabbit polyclonal antibody ChIP grade | Diagenode | C15410003-50 |
| Rabbit IgG isotype control | Invitrogen | 02-6102 |
| Parasite strains | | |
| NF54 *pfpeg4*-tdTomato | McLean et al (2018) | BKP_052 |
| Chemicals, peptides, and recombinant proteins | | |
| HEPES | Millipore | 391338 |
| RPMI 1640 | Corning | 50-020-PC |
| RMPI 1640 without phenol red | Corning | 90-022-PB |
| AlbuMAX II | Gibco | 11021-037 |
| Gentamicin | Gibco | 15710-064 |
| Hypoxanthine | Sigma-Aldrich | H9377 |
| Sodium bicarbonate | Sigma-Aldrich | 792519 |
| D-Sorbitol | VWR | 0691 |
| Percoll | Cytiva | 17089101 |
| Potassium chloride (KCl) | EMD | P×1405-1 |
| Calcium chloride (CaCl$_2$) | Sigma-Aldrich | C5670 |
| Manganese chloride (MnCl$_2$) | Acros Organic | 7773-01-5 |
| Sodium hydroxide (NaOH) | Sigma-Aldrich | S5881 |
| Potassium hydroxide (KOH) | Sigma-Aldrich | P-5958 |
| Sodium chloride (NaCl) | Sigma-Aldrich | S98888 |
| EGTA | Sigma-Aldrich | E0396 |
| Tris(hydroxymethyl)aminomethane | Roche | 11814273001 |
| Ethylenediaminetetraacetic acid dihydrate (EDTA) | VWR | 0105 |
| SDS | VWR | 0227 |
| Spermidine | Sigma-Aldrich | 85558 |
| DMSO | Corning | 25-950-CQC |
| RNase A, pancreatic | VWR | 97062-172 |
| PBS 10× pH 7.4 | Gibco | 70011-044 |
| 200 proof pure ethanol | Koptec | V1016 |
| Isopropanol | | |
| Digitonin, high purity | Millipore | 300410 |
| Propidium iodide | Sigma-Aldrich | P4864 |
| Hoechst 33342 | Invitrogen | H1399 |
| Thiazole orange | Sigma-Aldrich | 390062 |
| Proteinase K, PCR grade | Millipore-Sigma | 3115887001 |
| cOmplete, EDTA-free Protease Inhibitor Cocktail | Millipore-Sigma | 11836170001 |
| BioMag Plus Concanavalin A beads (5 mg/ml) | Bangs Laboratories | BP531 |
| Phenol chloroform-isoamyl alcohol 25:24:1 (PCI) | Sigma-Aldrich | 77617 |

**Continued**

| Reagent or resource | Source | Identifier |
|---|---|---|
| Chloroform | Calbiochem | 3150 |
| CUTANA pAG-MNase for ChIC/CUT&RUN | EpiCypher | 15-1016 |
| KAPA HIFI HotStart ready mix | Roche | 07958935001 |
| AMPure XP beads | Beckman Coulter | A63881 |
| Glycogen for molecular biology; Roche | Millipore-Sigma | 10901393001 |
| Critical commercial assays | | |
| NEBNext Ultra II DNA Library Prep kit for Illumina | NEB | E7645S |
| NEXTFLEX Unique Dual Index Barcodes (set D) | Bioo Scientific/Perkin Elmer | NOVA-514153 |
| Qubit dsDNA HS Assay Kits | Life Technologies | Q32854 |
| Oligonucleotides | | |
| Primer mix NEXTFLEX PCR primer 1: AATGATACGGCGACC ACCGAGATCTACAC PCR primer 2: CAAGCAGAAGACG GCATACGAGAT | Bioo Scientific/Perkin Elmer | NOVA-514153 |
| NEXTFLEX Unique Dual Index Barcode: AATGATACGGCGACCA CCGAGATCTACACXXXXXXXX[1]ACACTCTTTCCCTACACG ACGCTCTTCCGATCTGAT CGGAAGAGCACACGTCTG AACTCCAGTCACXXXXX XXX[2]ATCTCGTATGCCG TCTTCTGCTTG | Bioo Scientific/Perkin Elmer | NOVA-514153 |
| [1,2]Denotes P5 and P7 index region of adapter | | |
| Other | | |
| Veriti 96-well thermocycler | Applied Biosystems | 4375305 |
| Leica DMI6000 microscope | | |
| IncuBlock Plus heat block | Denville Scientific | I0601 |
| Microcentrifuge | Eppendorf | 5418 |
| Multi-rotator | Grant-bio | PTR-35 |
| Qubit 3.0 Fluorometer | Life Technologies | Q33216 |
| TapeStation System | Agilent | 4200 |
| 1.5-ml LoBind microcentrifuge tubes | Eppendorf | 022431021 |
| 0.2 ml PCR 8-tube strips | USA scientific | 1402-4700 |
| T25 flasks | VWR | 15708-120 |
| Transfer pipettes | VWR | 414004-002 |
| Magnetic bead separator stand | Invitrogen | F19900-0002 |
| 15 ml conical tubes | Corning | 430790 |
| 50 ml conical tubes | Corning | 430828 |
| Mr. Frosty Freezing Container | Thermo Fisher Scientific | 5100-0001 |
| Cold room | | |

## Solutions and buffers

### Binding buffer.

|  | Final concentration | Amount |
| --- | --- | --- |
| HEPES-KOH, pH 7.5 (1 M) | 20 mM | 400 $\mu$l |
| KCl (1 M) | 10 mM | 200 $\mu$l |
| CaCl$_2$ (100 mM) | 1 mM | 200 $\mu$l |
| MnCl$_2$ (100 mM) | 1 mM | 200 $\mu$l |
| ddH$_2$O | n/a | 19 ml |
| Total | n/a | 20 ml |

Store at 4°C for up to 6 mo. Note that the binding buffer uses HEPES-KOH!

### Wash buffer.

|  | Final concentration | Amount |
| --- | --- | --- |
| HEPES-NaOH, pH 7.5 (1 M) | 20 mM | 1 ml |
| NaCl (5 M) | 150 mM | 1.5 ml |
| Spermidine (2 M) | 0.5 mM | 12.5 $\mu$l |
| ddH$_2$O | n/a | 45.5 ml |
| Total | n/a | 50 ml |

Store at 4°C for up to 1 wk.
Critical: Protease Inhibitor Cocktail (EDTA-free) should be added fresh on the day of use.
Add 20 $\mu$l of Protease Inhibitor Cocktail stock solution per 1 ml of wash buffer.

### Dig-wash buffer.

|  | Final concentration | Amount |
| --- | --- | --- |
| Wash buffer | n/a | 30 ml |
| Digitonin (5%) | 0.025% (see note) | 150 $\mu$l |
| Total | n/a | 30 ml |

Store on ice or at 4°C for up to 1 d.
Critical: mix well before each use.
Note: digitonin has batch variability, and the minimal concentration that yields full of permeabilization needs to be determined empirically for your digitonin stock. We have obtained good results with as low as 0.025% digitonin. To test for permeabilization, pellet 150 $\mu$l of culture and resuspend in 150 $\mu$l Dig-wash buffer containing 8 $\mu$M Hoechst 33342 and 10 $\mu$g/ml propidium iodide. Incubate cell suspension for 10 min at RT and wash once. Pellet and resuspend cells in 20 $\mu$l of media and observe under fluorescence microscope or flow cytometer. Determine the lowest digitonin concentration at which all nuclei are positive for both dyes (Fig 5).

### Antibody buffer.

|  | Final concentration | Amount |
| --- | --- | --- |
| Dig-wash buffer | n/a | 2 ml |
| EDTA, pH 8.0 (0.5 M) | 2 mM | 8 $\mu$l |
| Total | n/a | 2 ml |

Always prepare fresh before use.
Critical: add the amount of antibody indicated right before incubation and keep on ice at all times.

### Low-salt rinse buffer.

|  | Final concentration | Amount |
| --- | --- | --- |
| HEPES-NaOH, pH 7.5 (1 M) | 20 mM | 400 $\mu$l |
| Digitonin (5%) | 0.025% | 100 $\mu$l |
| Spermidine (2 M) | 0.5 mM | 12.5 $\mu$l |
| ddH$_2$O | n/a | 19.5 ml |
| Total | n/a | 20 ml |

Store at 4°C for up to 1 wk.
Critical: Protease Inhibitor Cocktail (EDTA-free) should be added fresh on the day of use.
Add 20 $\mu$l of 25× stock solution per 1 ml of buffer.

### High Ca++ incubation buffer.

|  | Final concentration | Amount |
| --- | --- | --- |
| HEPES-NaOH, pH 7.5 (1 M) | 3.5 mM | 14 $\mu$l |
| CaCl2 (100 mM) | 10 mM | 400 $\mu$l |
| Digitonin (5%) | 0.025% | 20 $\mu$l |
| ddH$_2$O | n/a | 3.6 ml |
| Total | n/a | 4 ml |

Store at 4°C for up to 1 wk.
Critical: Protease Inhibitor Cocktail (EDTA-free) should be added fresh on the day of use.
Add 20 $\mu$l of 25× stock solution per 1 ml of buffer.

### Stop buffer.

|  | Final concentration | Amount |
| --- | --- | --- |
| NaCl (5 M) | 170 mM | 170 $\mu$l |
| EGTA (100 mM) | 20 mM | 1 ml |
| Digitonin (5%) | 0.025% | 25 $\mu$l |
| RNase A (10 mg/ml) | 50 $\mu$g/ml | 12.5 $\mu$l |
| Glycogen (20 mg/ml) | 25 $\mu$g/ml | 6.25 $\mu$l |
| ddH$_2$O | n/a | 3.8 ml |
| Total | n/a | 5 ml |

Store at 4°C for up to 1 wk.

**TE buffer.**

|  | Final concentration | Amount |
|---|---|---|
| Tris–HCl (1 M) | 10 mM | 1 ml |
| EDTA (500 mM) | 1 mM | 0.2 ml |
| ddH$_2$O | n/a | 98.8 ml |
| Total | n/a | 100 ml |

**Step-by-step instructions**

### Activation of concanavalin A-coated beads

**Timing: 5 min** Concanavalin A lectins on magnetic beads are activated and stabilized by Ca$^{2+}$ and Mn$^{2+}$ cations.

(1) Gently resuspend and withdraw 28 $\mu$l of the ConA bead slurry per sample/antibody combination.
(2) Transfer beads to a 1.5-ml microcentrifuge tube and add 1 ml of binding buffer.
(3) Place the tube on a magnetic bead separator stand and allow to clear (30–120 s) before removing and discarding the supernatant.
(4) Remove tube from stand and add 1 ml of binding buffer. Mix gently and return to the magnetic stand.
(5) Repeat steps 3–4 one additional time.
(6) Resuspend beads in the same volume of binding buffer that was added in step 1 and keep at RT.
(7) Place wash buffer at RT to warm for later use.

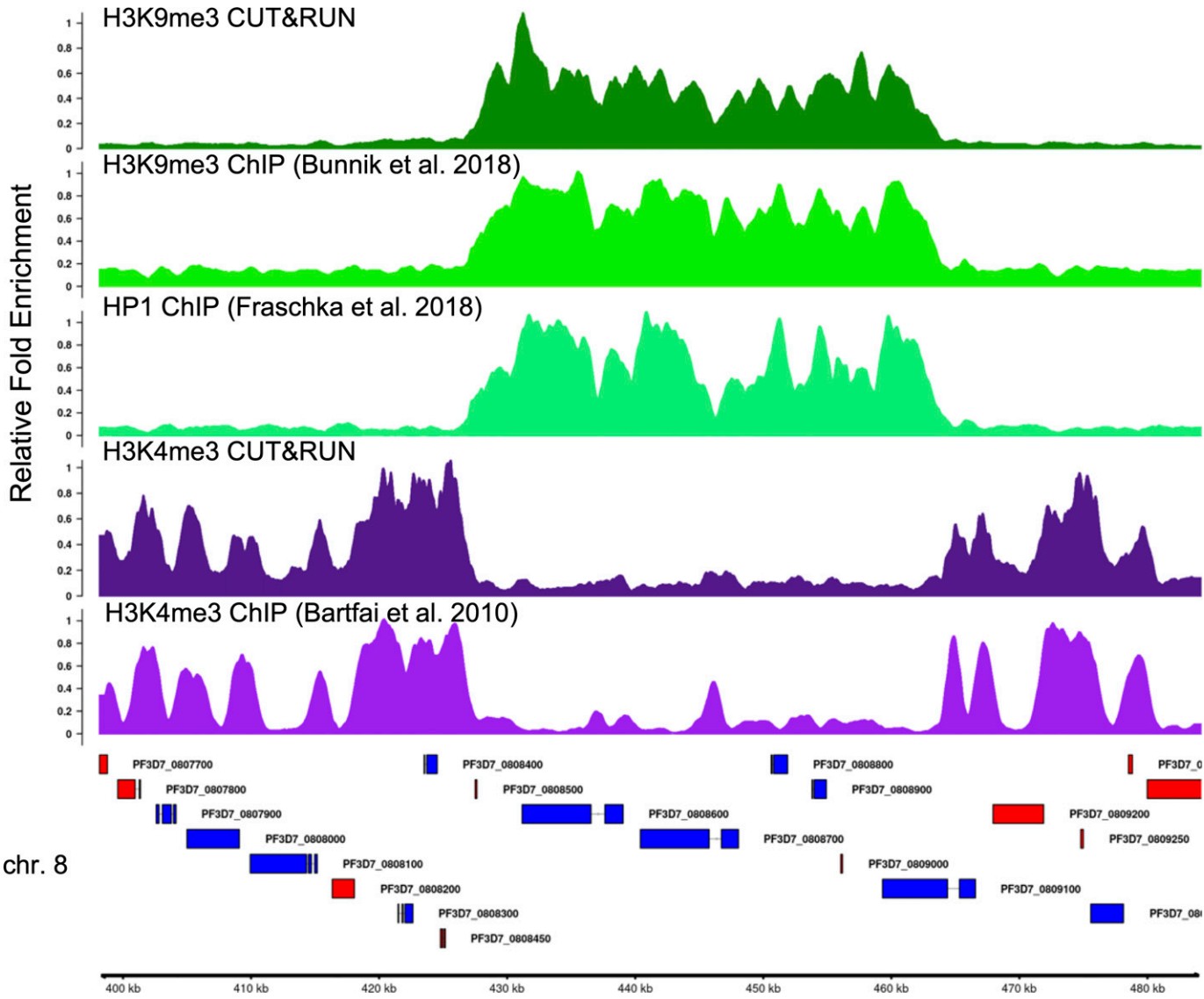

**Figure 3. Comparison of H3K4me3 and H3K9me3 profiles obtained by CUT&RUN to published ChIP-seq profiles at an example locus containing a non-subtelomeric heterochromatin island on chromosome 8.**
Tracks show the relative fold enrichment of specific histone modifications, H3K9me3 and H3K4me3, and for heterochromatin protein 1, the H3K9me3 histone reader, versus either isotype controls for CUT&RUN or input DNA (ChIP-seq). Sequence reads from published stage-matched H3K9me3 (SRR4444647, SRR4444639), HP1 (SRR5935737, SRR5935738), and H3K4me3 (SRR065659, SRR065664) ChIP-seq experiments were downloaded from the NCBI Sequence Read Archive. Genes shown in blue are encoded on the top (+) strand, whereas those in red are encoded on the bottom (−) strand. Gene IDs are shown to the right of each gene.

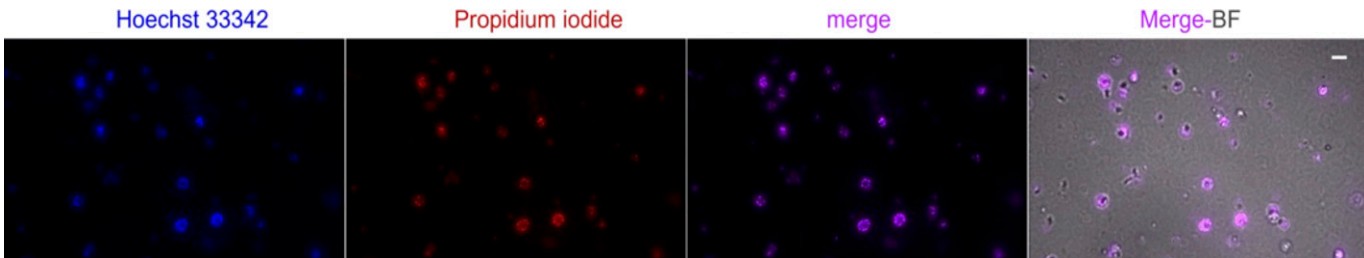

**Figure 4.   Fold Enrichment profiles of down-sampled H3K9me3 (green) and H3K4me3 (purple) CUT&RUN reads show good signal-to-noise ratios down to ~1 million read-pairs per sample.**

Read pairs were serially down-sampled in silico by fourfold to evaluate the signal-to-noise ratio. Each track is scaled to its maximal value across the locus shown. The number of sequenced paired-end 50 bp (PE50) clusters used is indicated in millions. Example locus on chromosome 8 contains a non-subtelomeric heterochromatin island. Genes shown in blue are encoded on the top (+) strand, whereas those in red are encoded on the bottom (−) strand. Gene IDs are shown the above or below the gene. Grey regions in the 64× down-sampled tracks lack coverage and are inferred based on neighboring bins.

**Figure 5.   iRBCs treated with 0.025% digitonin and stained with Hoechst 33342 and propidium iodide.**

Nuclei of permeabilized cells will stain with both dyes. Scale bar, 4 μm.

### Binding cells to concanavalin A-coated beads

**Timing: 1 h** Infected red blood cells (iRBCs) bind to beads coated with concanavalin A, a lectin that binds specifically to extracellular glycoproteins.

(8) Harvest fresh parasite cultures at the desired stage and synchrony. If feasible, enrich for iRBCs by magnetic purification (Mata-Cantero et al, 2014), centrifugation in a continuous percoll gradient (Rivadeneira et al, 1983), or lysis of uninfected RBCs (Brown et al, 2020). The stage, synchrony, and growth conditions are determined by the scientific question the experimenter is asking. In this study, we used percoll/sorbitol isolated asexual blood stages synchronized to a window of 36 ± 4 h post-invasion.

(9) Determine iRBC density and purity using a hemocytometer, coulter counter, or volumetric flow cytometry.

(10) Pellet cells by centrifugation (800$g$ for 3 min at RT) and resuspend cells containing 1–5 × 10$^7$ nuclei to a cell density to 1 × 10$^7$ cells/ml in wash buffer by gentle pipetting.

Note: the average nuclear content of parasites can be determined by flow cytometry after staining cultures with 8 $\mu$M Hoechst 33342 (DNA specific) and 0.1 $\mu$g/ml thiazole orange (RNA specific with Hoechst 33342) for 30 min at 37°C, followed by two brief washes. A high ring-stage culture can be used to determine the mean fluorescence value equivalent to a 1 N nuclear content (Fig 6). However, we did not attempt to optimize the minimal number of nuclei required. Based on CUT&RUN in other systems and our down-sampling analysis (Fig 4), substantially fewer nuclei are likely sufficient.

Optional: at this point, it is possible to cryopreserve infected RBCs for later processing. Harvest cells by centrifugation (800$g$ for 3 min at RT) and resuspend in 1 ml of 10% DMSO prepared in 1× PBS and freeze slowly to –80° C using an isopropanol freezing chamber to minimize cellular damage. Store frozen cells at –80°C until you are ready to use the cells, then thaw them quickly by placing cryovials in a 37°C water bath for 3–5 min. Then transfer the cell suspension to a 1.5-ml microcentrifuge tube and spin at 800$g$ for 3 min at RT. Wash twice with 1× PBS before proceeding to step 13.

Critical: from here on, mix gently and avoid vortexing.

(11) Split each sample into 1 ml aliquots (one per antibody including isotype controls) in 1.5-ml LoBind DNA microcentrifuge tubes and wash twice with 1 ml of wash buffer (800$g$ for 3 min at RT).

(12) Resuspend cells in 225 $\mu$l wash buffer by gently pipetting.

(13) Add 25 $\mu$l of activated concanavalin A-coated magnetic bead slurry to each tube.

**Figure 6. Quantification of parasite genome content.**
Left: flow cytometric analysis of live cultures after staining for 30 min with Hoechst 33342 and thiazole orange allows easy differentiation of uninfected RBCs (red), ring-stage infected cells (blue), and late-stage infected cells (orange) based on RNA and DNA content. Right: 1N ring stages can be used as the reference for calculating genome content of late stages.

(14) Incubate at RT on rotator for 10 min at an orbital speed of 10 rpm.

Note: 125 μg of beads (the equivalent of 25 μl of bead slurry) in a 250 μl volume captured 90% of $1 \times 10^6$ and $3 \times 10^6$ cells or 60% of $1 \times 10^7$ cells in 10 min.

### Cell permeabilization and antibody binding

**Timing: 2.5 h** Cells attached to conA beads are permeabilized to allow antibodies to diffuse into the nucleus and bind their targets.

(15) For each antibody, make an antibody master mix by combining antibody buffer with 0.8 μg of antibody to a final volume of 165 μl per sample. For antibodies other than those used in this protocol, final concentrations may have to be determined empirically based on affinity and target abundance.

(16) Ensure bead-bound cells are in homogenous suspension by inverting the tube several times, then divide bead-bound cells into aliquots of 250 μl in 1.5-ml microcentrifuge LoBbind DNA tubes.

Note: if antibodies from multiple species are used, include the appropriate isotype control for each species. The amount of antibody needed for optimal signal depends on multiple factors, including its affinity for the target, the abundance of the target, and the number of cells used. In our experience, high background is more often caused by using too much antibody rather than too little.

(17) Place tubes on a magnetic stand to clear, then carefully remove and discard supernatant.

(18) Remove the tube from the stand and, while holding the tube at an angle, add 150 μl of antibody master mix to the side of the tube opposite from the beads. Then tap gently to dislodge the beads.

(19) Incubate on a rotator for 2 h at an orbital speed of 10 rpm at RT.

Pause point: antibody incubation may proceed overnight at 4°C.

### Secondary antibody binding (optional)

**Timing: 1.5 h** Note: the affinity of protein A/G to IgG antibodies varies based on host species and IgG subtype (www.bio-rad.com/en-us/applications-technologies/protein-g-affinity). If mouse primary antibodies are being used, it is recommended to use a secondary rabbit-anti-mouse antibody to increase binding to protein A. Although the version of pA/G–MNase sold by Epicypher has an improved antibody compatibility relative to protein A alone, Jenssen and Henikoff suggest that rabbit anti-mouse secondary antibody should still be used in some cases (Janssens & Henikoff, 2019 Preprint). Otherwise, proceed to step 26.

(20) For each mouse antibody, make a secondary antibody master mix by combining antibody buffer with 0.8 μg of rabbit anti-mouse IgG to a final volume of 165 μl per sample with mouse primary antibody. Remove liquid from cap and side

with a quick pulse on a micro-centrifuge, then place on the magnetic stand to clear then remove and discard the supernatant.

(21) Add 200 μl of Dig-wash buffer, mix by pipetting gently and place on the magnetic stand to clear before removing and discarding the supernatant. Repeat this step two more times.

(22) Hold the tube at an angle and add 150 μl of secondary antibody master mix to the opposite side of the tube from the cells. Tap gently to dislodge the beads.

(23) Place the tube on a rotator at 4°C for 1 h at 10 rpm orbital speed.

### Protein A/G–MNase binding

**Timing: 1 h** Protein A/G-MNase fusion proteins diffuse into the cells and complex with the chromatin-bound antibodies.

(24) Remove liquid from cap and side of the tube with a quick pulse on a micro-centrifuge, then place on the magnetic stand to clear and pull off all the liquid.

(25) Add 200 μl of Dig-wash buffer, mix by pipetting gently, then place on the magnetic stand to clear and remove all the liquid. Repeat this step two more times.

(26) Holding the tube at an angle, add 150 μl of the protein A/G–MNase fusion protein in Dig-wash buffer (2.5 μl of 20× stock per sample) directly onto the beads. Tap to dislodge the remaining beads.

(27) Place the tube on a rotator at 4°C for 1 h at 10 rpm orbital speed.

(28) Place incubation buffer and an aluminum block for 1.5-ml tubes on ice for later use.

### Chromatin digestion and release

**Timing: 1 h** Chromatin-associated MNase is activated by adding high $Ca^{2+}$ incubation buffer. Digestion at 0°C prevents the premature diffusion of MNase/DNA complexes and reduces background.

(29) Remove liquid from cap and side with a quick pulse on a microcentrifuge, then place on the magnetic stand to clear, and discard supernatant.

(30) Add 200 μl of Dig-wash buffer, mix by pipetting gently, then place on the magnetic stand to clear, and discard supernatant. Repeat this step two more times.

(31) Add 200 μl of low-salt rinse buffer, mix by pipetting gently, then place on the magnetic stand to clear, and discard supernatant.

(32) Add 200 μl of ice-cold incubation buffer directly onto the beads. Tap to dislodge the remaining beads.

(33) Incubating at 0°C for 30 min using the pre-chilled aluminum heater block to allow MNase to release bound nucleosomes. Block should be at 0°C to minimize background cleavage.

(34) Place on magnetic stand at 4°C, allow to clear, then remove, and save the supernatant for possible troubleshooting (see the Troubleshooting section at the end).

(35) Add 200 μl stop buffer to beads and mix by pipetting gently.

(36) Allow liberated nucleosomes to diffuse from the nucleus into the supernatant by incubating for 30 min in a 37°C water bath. Critical: do not rotate!

(37) Place tube on the magnetic stand to clear. Without disturbing the beads, carefully transfer the supernatant containing liberated chromatin to a fresh LoBind DNA tube.

### DNA purification by phenol/chloroform extraction

**Timing: 2 h** After the release and diffusion of the nucleosome fragments into the supernatant, DNA is purified and subjected to quality control.

(38) To each sample (200 $\mu$l of supernatant containing released CUT&RUN fragments), add 2 $\mu$l 10% SDS and 2.5 $\mu$l roteinase K (20 mg/ml). Mix by inversion and incubate for 1 h at 50°C in a water bath.

(39) Add 205 $\mu$l of phenol-chloroform-isoamyl alcohol 25:24:1. Mix by vortexing at full speed for 5 s.

(40) Spin at 16,000$g$ for 5 min at RT.

(41) Without disturbing the interface, transfer the top liquid phase to a fresh LoBind tube.

(42) Add an equivalent volume of chloroform, invert 10 times to mix, then spin at 16,000$g$ for 5 min at RT.

(43) Transfer the top liquid phase to a fresh LoBind tube and add 2 $\mu$l of 2 mg/ml glycogen.

(44) Add 500 $\mu$l of 100% ethanol and mix 10× by inversion. With an ethanol-resistant pen, mark the bottom of the tube on one side where the pellet is going to form. Chill on ice for 10 min then spin at 16,000$g$ for 10 min at 4°C with the mark facing outward.

(45) Carefully pour off the liquid and dry tube rim by touching it to a paper towel. Rinse the pellet in 1 ml of 100% ethanol and spin at 16,000$g$ for 10 min at 4°C.

Note: the pellet may not be visible at this point, so be careful to not discard the pellet.

(46) Carefully pour off the liquid and dry on a paper towel. Let tube air dry on its side for 5 min.

(47) When the pellet is dry, dissolve in 30 $\mu$l TE buffer.

(48) Take 2 $\mu$l of each sample to quantify DNA concentration by Qubit Fluorometer using the Qubit dsDNA HS (High Sensitivity) assay kit. In our experience, concentrations at this step are nearly always below 1 ng/$\mu$l and have little predictive value of downstream success.

(49) Keep the samples on ice at all times.

Note: although samples can be stored at −20°C, it is strongly advisable to continue with the protocol until libraries have been PCR amplified.

### End repair and A-tailing

**Timing: 2 h** The ends of the extracted DNA are repaired and A-tailed to allow subsequent HTS adapter ligation.

(50) Place AMPure XP Beads at RT for later use.

(51) In a 1.5-ml microcentrifuge tube, prepare the End repair master mix.

**End repair master mix.**

| Reagent | Amount per reaction* |
| --- | --- |
| EndPrep Enzyme Mix | 1.65 $\mu$l |
| EndPrep buffer | 3.85 $\mu$l |

*Master mix recipes include 10% extra volume for pipetting error.

(52) Combine 5 $\mu$l of end repair master mix to PCR tubes with 4 ng of DNA template and add ddH$_2$O to a total volume of 30 $\mu$l. Mix by vortexing and quick spin to collect at the bottom of the tube.

(53) Place the tube in a PCR machine with the heated lid set to ≥75°C and run the following program.

**End repair and A-tailing reaction conditions.**

| Step | Temperature | Time | Cycles |
| --- | --- | --- | --- |
| End repair | 20°C | 30 min | 1 |
| A-tailing | 50°C | 60 min | 1 |
| Hold | 4°C | forever | |

Note: compared with manufacturer instructions, the A-tailing temperature was decreased to 50°C to avoid DNA melting of shorter DNA fragments and the reaction time was increased to 60 min to compensate for lower enzymatic efficiency.

### Ligation of barcoded HTS adapters

**Timing: 25 min** Dual barcode HTS adapters are added to the end-repaired and A-tailed fragments.

(54) Remove tube from PCR machine and place on ice.

(55) In a 1.5-ml microcentrifuge tube, combine the following reagents.

**Adapter ligation master mix.**

| Reagent | Amount per reaction |
| --- | --- |
| NEB ligation mix | 18 $\mu$l |
| NEB ligation enhancer | 0.6 $\mu$l |
| NEXTFLEX Unique Dual Index Barcode adapters at 0.05 $\mu$M | 1.8 $\mu$l |

Note: we use adapters from NEXTFLEX Unique Dual Index Barcodes (25 $\mu$M concentration) diluted 1:500 in ddH$_2$O (0.05 $\mu$M final stock concentration). Because adapters are supplied in duplex form, avoid heating above RT. Dual index barcodes should be used when sequencing on Illumina systems that use patterned flow cells. If in doubt, use dual index barcodes.

(56) Add 17 $\mu$l of Adapter ligation master mix (final volume of 47 $\mu$l) and mix by vortexing. Quick spin to collect liquid at the bottom of the tube.

(57) Place the tube in the PCR machine with the heated lid OFF and incubate for 15 min at 20°C.

### Post-ligation clean-up
**Timing: 20 min** Excess HTS adapters are removed.

(58) Transfer the samples into a 1.5-ml LoBind microcentrifuge tube.
(59) Add 80 µl of AMPure XP bead slurry (1.7× ratio). Mix well by pipetting at least 10 times and incubate for 5 min at RT.

Note: before use, equilibrate AMPure XP beads to RT for at least 30 min.

(60) Place the tube on the magnetic stand to separate beads from liquid. Discard the supernatant.
(61) Wash twice with 200 µl 80% ethanol while keeping the tubes on the magnetic stand.
(62) Discard supernatant and let dry for up to 5 min

Critical: do not discard the beads! Tubes must remain in the magnetic stand the whole time. Avoid over-drying the beads. Add TE buffer to the beads when all visible liquid has evaporated, but beads are still dark brown and glossy looking.

(63) Resuspend beads in 15 µl of TE buffer. Mix by pipetting 10 times, and incubate for 5 min at RT.
(64) Place the tube back on the magnetic stand. Allow to clear and transfer the supernatant (~13 µl) to a fresh PCR tube.

### Library PCR amplification
**Timing: 35 min** Fragments are PCR amplified using KAPA polymerase, which is more efficient at amplifying AT-rich fragments.

(65) In a 1.5-ml microcentrifuge tube, combine the following reagents.

**PCR master mix.**

| Reagent | Amount per reaction |
| --- | --- |
| 2× KAPA HotStart ready mix | 27.5 µl |
| NEXTFLEX Primer mix | 5.5 µl |
| ddH$_2$O | 7.7 µl |

Note: we use NEXTFLEX Primer Mix (12.5 µM stock concentration) that comes with NEXTFLEX Unique Dual Index barcodes sets, at a final concentration of 1.2 µM.

(66) Add 37 µl of PCR master mix to each sample for a final volume of 50 µl. Mix by vortexing and then quick spin to collect liquid at the bottom of the tube.
(67) Place in a PCR machine with the heated lid set to 100°C and run the following program.

**PCR cycling conditions.**

| Step | Temperature | Time | Cycles |
| --- | --- | --- | --- |
| Initial denaturation | 98°C | 60 s | 1 |
| Denaturation | 98°C | 10 s | 15 cycles |
| Annealing and extension | 65°C | 60 s | |
| Final extension | 65°C | 5 min | 1 |
| Hold | 4°C | forever | |

Pause point: samples can be stored at –20°C until further use.

### Clean-up of PCR amplified library
**Timing: 35 min** Excess primers are removed.
Note: this round of selection is aimed to remove primer and adapter pairs below 150 bp.

(68) Add 50 µl of AMPure XP bead slurry (1.0× ratio) to each amplified library and mix by pipetting at least 10 times, then incubate for 5 min at RT.
(69) Place the tube on the magnetic stand for 5 min to separate beads from supernatant.
(70) After the solution is clear, carefully remove and discard the supernatant. Be careful not to disturb the beads that contain DNA targets.

Critical: do not discard the beads! Tubes must remain in the magnetic stand the whole time.

(71) Add 200 µl of 80% ethanol (freshly prepared) to the tube while in the magnetic stand. Incubate at RT for 30 s, and carefully discard the supernatant. Repeat this step for a total of two washes, and remove all visible liquid after the second wash.
(72) Air dry the beads for up to 5 min while the tube is on the magnetic stand with the lid open. Avoid over-drying the beads.
(73) Remove the tube from magnetic stand. Elute the DNA fragment from the beads by adding 15 µl of TE buffer. Mix well by pipetting up and down 10 times. Incubate for 2 min or more at RT.
(74) Place the tube on the magnetic stand. After 5 min (or when the solution is clear), transfer the supernatant (~13 µl) to a new fresh microcentrifuge LoBind tube.

Pause point: samples can be stored at –20°C until further use.

### Library quality control and Illumina sequencing
**Timing: 25 min**
(75) Take 2 µl of each sample to quantify DNA concentration by using fluorescence detection Qubit Fluorometer using the Qubit dsDNA HS (High Sensitivity) assay kit. For the antibodies/samples in this example, concentrations were usually in the 2–15 ng/µl range (lower for isotype controls).
(76) Use 1 ng of each library to check fragment size distribution and molar concentration using a Bioanalyzer or TapeStation system (See Fig 7).

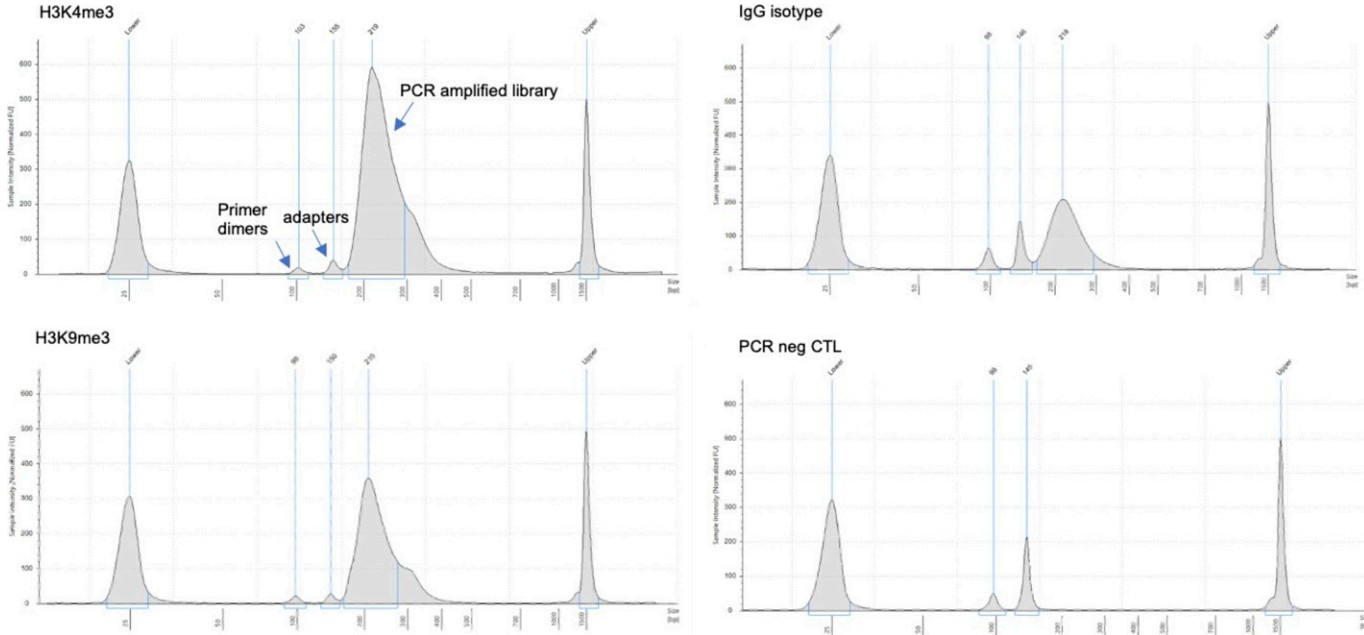

**Figure 7.  Example size distributions of CUT&RUN libraries from H3K9me3, H3K4me3, IgG isotype, and PCR negative control as analyzed by TapeStation.**
Primer dimer, adapter, and library peaks are indicated in the top left panel.

Note: if the fraction of adapter dimers exceeds 20% of your library, an additional round of size selection is required. Do not pool libraries with high adapter content with high-quality libraries as this will require size selecting the pool and reduce the abundance of high-quality libraries.

(77) Pool libraries at desired ratio reflecting the desired relative sequencing depths for each library (in this case, we combined equimolar amounts (0.02 pmol) of 25 libraries to generate the library pool).
(78) Rerun pooled libraries in a Bioanalyzer or TapeStation to check size profile and molar concentration.
(79) Perform paired-end Illumina sequencing following manufacturer's instructions. 50 bp paired-end reads are sufficient for accurate mapping.

## Expected outcomes

When starting with $1 \times 10^7$ schizont stages with an average nuclear content of 3–5 N, post-clean-up library yield was in the range of 10–40 ng after 15 cycles of PCR. Fig 2 shows examples of CUT&RUN library size profiles obtained with anti-H3K9me3 and anti-H3K4me3 antibodies. The ratio of primer dimers and adapters in our libraries was around 4%, which is 5× lower than normally reported for these assays. Library size profile obtained with rabbit IgG antibody represents unspecific bound regions that have been PCR amplified. Also, a PCR negative control with no template DNA was shown to dismiss this sort of contamination during library preparation.

## Quantification and statistical analysis

The libraries shown in these examples were pooled with other CUT&RUN libraries at equimolar ratios and sequenced using an Illumina NextSeq2000 sequencer in paired-end 50 bp P2 mode. We initially aimed at a target of 10 million PE50 read pairs per library to ensure good coverage. Subsequent down-sampling analysis showed that as low as 1 million read pairs provide excellent signal-to-noise ratios for these antibodies (Fig 4).

For the comparison of the CUT&RUN profiles to published profiles (Fig 3), we obtained raw read files from public repositories using SRAtools (https://github.com/ncbi/sra-tools). H3K9me3 and input raw ChIP-seq reads from Bunnik et al (2018) were downloaded from NCBI Sequence Read Archive (accession number SRP091939). PfHP1 and input raw ChIP-seq reads from Fraschka et al (2018) were downloaded from NCBI's Gene Expression Omnibus (GEO) (accession number GSE102695). H3K4me3 and input raw ChIP-seq reads from Bártfai et al (2010) were downloaded from NCBI's GEO (accession number GSE210062).

Raw reads were quality trimmed using Trimmomatic v0.38 (Bolger et al, 2014) to remove residual adapter sequences and low-quality leading and trailing bases to retain properly paired reads of length ≥ 30 bases after trimming, as previously published (Campelo Morillo et al, 2022). Trimmed reads were aligned to PlasmoDB version 51 *P. falciparum* 3D7 reference genome (Warrenfeltz et al, 2018) using BWA v.0.7.1 (Li & Durbin, 2009). SAMtools v.1.10 (Li et al, 2009) was used to sort and index BAM files. Normalized fold enrichment tracks were generated by using the MACS2 v.2.2.7.1 (Zhang et al, 2008) callpeak function with settings: -f BAMPE -B -g $2.3 \times 10^7$ -q 0.05 –nomodel–broad–keep-dup auto–max gap 500. BedGraph outputs were then passed into the bdgcmp function with the setting -m FE (fold enrichment) to generate signal tracks to profile histone modification enrichment levels compared with either input (for ChIP-seq) or isotype IgG control (CUT&RUN). Using the bedGraphToBigWig tool (Kent et al, 2010), we converted the genome-wide coverage and enrichment tracks from bedGraph into BigWig

format. These files are available for download at NCBI GEO under accession number GSE210062 and can be directly loaded into the PlasmoDB genome browser (Warrenfeltz et al, 2018) for detailed inspection.

Coverage and enrichment tracks were visualized using Integrative Genome Viewer IGV (Thorvaldsdóttir et al, 2013) or the GenomicRanges v.1.44.0 and GViz Viz v1.38.1 (Hahne & Ivanek, 2016) R packages within the Bioconductor project (release 3.13) (Huber et al, 2015). The full analysis pipeline can be found at https://github.com/KafsackLab/PfCUTandRUN. Representative tracks of H3K9me3, H3K4me3, and HP1 produced using CUT&RUN or ChIP-seq at chromosome 8 of *P. falciparum* are shown in Fig 4.

### Troubleshooting

***Problem 1: low DNA concentration after library amplification***
If DNA is low or undetectable in step 84, check the following.

-Beads were not activated properly.
-Beads got stuck on tube lid or walls during incubation steps.
-Diffusion of MNase/DNA/antibody complex occurred prematurely.

**Potential solution** Remember to activate beads before initiating the protocol. Give a quick spin after each incubation to recover beads from tube lid and walls. Try not to pipette too harshly because cells can be very fragile after permeabilization. Always perform the digestion at 0°C, and do not agitate the tube once high $Ca^{2+}$ incubation buffer is added to the isolated DNA from supernatants saved in step 36. No DNA should be detected in this fraction; if so, this is indicative of premature diffusion of nucleosomes during digestion. For low-abundance targets, increasing the cycle number during PCR library amplification may help.

### Limitations

This protocol relies on the availability of a specific antibody against histones PTM that would be able to recognize correspondent modification in *P. falciparum*. This should be tested empirically by Western blot or immunoprecipitation assays. We have only used rabbit antibodies. Antibodies generated in other species should be tested according to experimental set up and availability, and antibodies raised in other species may benefit from the inclusion of rabbit secondary antibodies. This protocol was optimized for use with abundant chromatin marks on nucleosomes. For rare modifications, or non-nucleosomal targets, like transcription factors and histone modifiers, further optimization may be required.

## Data Availability

The full analysis pipeline can be found at https://github.com/KafsackLab/PfCUTandRUN. Raw (fastq) and processed (BigWig) high-throughput sequencing data have been deposited in the NCBI GEO under accession number GSE210062.

## Supplementary Information

## Acknowledgements

We thank the VEuPathDB team and the Weill Cornell Medicine genomics core for technical support. This work was supported by funding from Weill Cornell Medicine, NIH NIAID 1R01 AI141965 and NIH NIAID 1R01 AI138499, Alice Bohmfalk Charitable Trust, and Charles A Frueauff Foundation.

### Author Contributions

RC Morillo: formal analysis, validation, investigation, visualization, methodology, and writing—original draft, review, and editing.
CT Harris: data curation, software, formal analysis, validation, investigation, visualization, methodology, and writing—original draft, review, and editing.
K Kennedy: visualization.
SR Henning: investigation.
BFC Kafsack: conceptualization, data curation, software, formal analysis, supervision, funding acquisition, visualization, methodology, project administration, and writing—review and editing.

### Conflict of Interest Statement

The authors declare that they have no conflicts of interest.

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
