## [Reviewer comments · Life Science Alliance]

Life Science Alliance

Genome-wide Profiling of Histone Modifications in *Plasmodium falciparum* using CUT&RUN.

Riward Morillo, Chantal Harris, Kit Kennedy, Samuel Henning, and Bjorn Kafsack

DOI: <https://doi.org/10.26508/lsa.202201778>

Corresponding author(s): Bjorn Kafsack, Weill Cornell Medicine and Riward Morillo, Weill Cornell Medicine

Review Timeline:

Submission Date:	2022-10-20
Editorial Decision:	2022-10-27
Revision Received:	2022-10-27
Accepted:	2022-10-28

Transaction Report:

Please note that the manuscript was reviewed at Review Commons and these reports were taken into account in the decision-making process at Life Science Alliance.

Revision Plan

Manuscript number: #RC-2022-01617

Title: Genome-wide Profiling of Histone Modifications in *Plasmodium falciparum* using CUT&RUN

Corresponding author: Björn Kafsack,

1. General Statements

With three exceptions, we have incorporated all other reviewer suggestions in this revised manuscript. Their thoughtful comments were greatly appreciated in making this manuscript even more accessible to the wider readership.

2. Description of the planned revisions

Based on the reviewer comments, we don't feel that additional experiments are warranted.

3. Description of the revisions that have already been incorporated in the transferred manuscript

The reviewer's enthusiasm and thoughtful comments on how to improve the manuscript were greatly appreciated and their valuable suggestions were incorporated into this revised version of the manuscript. We have responded to each of the reviewer comments below.

Reviewer comments are in black. Our Responses are in Blue.

Reviewer #2 (Evidence, reproducibility and clarity (Required)):

This paper provides a detailed step by step protocol of the CUT&RUN technique, which enables high-resolution chromatin mapping and probing, adapted to the malaria parasite *Plasmodium falciparum*. In particular, Kafsack and colleagues apply the CUT&RUN protocol to infected red blood cells from in-vitro culture and obtain very good quality genome-wide profiles of two histone modifications, H3K4me3 and H3K9me3. The results are congruent with previous ChIP-seq data with a substantial improvement in terms of coverage and chip-to-input noise. The protocol is very detailed and the figures are great.

Major comments:

- Authors successfully adapted the CUT&RUN protocol in *P. falciparum*. First, the binding profiles obtained by CUT&RUN for H3K4me3 and H3K9me3 are very similar to those reported by previous ChIP-seq studies. Secondly, by down-sampling 4X and 16X the test samples, authors demonstrate that 1M PE reads of sequencing depth would be enough to obtain accurate profiling of these histone modifications.

Revision Plan

Despite this data is convincing, only one region in chr. 8 is shown as an example in figures 2, 3 and 4. Different regions should be included, at least as supplementary figures, to reinforce their conclusions. We chose that locus on chromosome 8 to provide a gene-level resolution view at a locus encompassing genes in both eu- and heterochromatin states. We have now included Supplementary Figure 1, which shows these tracks for full length chromosomes 4 and 7. Additionally, genome-wide enrichment tracks for all data sets in this study are available for download at NCBI Gene Expression Omnibus under accession number GSE210062.

Related to this, there is evidence of the impact of chromatin structure on ChIP-seq analysis. Specifically, heterochromatin is typically depleted in ChIP input controls because of technical and experimental issues and this can result in a false enrichment of heterochromatic regions in the tested sample. How represented is heterochromatin (i.e. sub-telomeric and telomeric regions) in the test and control samples using the cut&run protocol?

The reviewer is correct that chromatin structure may alter accessibility which may bias absolute measurements but since the accessibility biases based to chromatin-structure are identical for both the histone PTM-specific antibody and the isotype control and cancel out in the enrichment score.

How biased is the cut&run sample compare to the ChIP-seq sample?

We have included this in Supplementary Figure 1. The H3K4me3 and H3K9me3 enrichment scores are strongly correlated both between CUT&RUN replicates and between CUT&RUN and previously published ChIP-seq results.

In this sense, it would be desirable if authors provide more information about the quality analysis results, for example the chip to input signal ratio and the coverage for heterochromatic (telomeric, centromeric and subtelomeric) regions.

We agree that this would be of interest to the reader. For this reason, the full genome-wide enrichment tracks were made available for all datasets in this study. We have added language to draw further attention to this availability.

Additionally, loci typically biased in ChIP-seq samples, i.e. clonally variant gene families in sub-telomeric regions, should be shown as examples.

We chose that locus on chromosome 8 to provide a gene-level resolution view at a locus encompassing genes in both eu- and heterochromatin states, including 2 *var* genes (PF3D7_0808600 and PF3D7_0808700) and two *rifin* genes (PF3D7_0808800 and PF3D7_0808900). We have now included Supplementary Figure 2, which shows these tracks for full length chromosomes 4 and 7, which also include subtelomeric and non-subtelomeric heterochromatin loci containing these genes. Additionally, genome-wide enrichment tracks for all data sets in this study are available for download at NCBI Gene Expression Omnibus under accession number GSE210062.

Revision Plan

- The step by step protocol is very detailed, however there are some parts that need to be better explained:

In the section "Binding cells to Concanavalin A-coated beads": it's not mentioned the harvest time and the stage of the parasites used. In addition, several methods are proposed for iRBCs enrichment, but is not mentioned which method was used and the life stage of the parasites. In this part of the protocol authors state "resuspend cells containing $1-5 \times 10^7$ nuclei to a cell density to 1×10^7 cells/mL".

According to our calculations, to guarantee this nuclei number it would be necessary to enrich in iRBCs and late stages. Otherwise the red blood cells density should be much larger. Could you please clarify this point?

For this study we enriched for trophozoites using a percoll/sorbitol density gradient, which we have now clarified in step 8. However, whether and which enrichment strategy is employed will vary based on the desired parasite stages and experimental design.

In the section "P. falciparum culturing and synchronization of erythrocytic stages" the authors indicate that the method used for synchronization was double-synchronization with sorbitol treatment to achieve a {plus minus} 6 h synchrony. The details provided appear insufficient to replicate the procedure. E.g. it's not explained how the double step synchronization was performed and for how long the culture was incubated after the synchronization.

The number of parasite cells and the life-stage used is mentioned at the end (in the section of expected outcomes). It would be more useful if this information is specified at the beginning together with the most appropriate procedure to get an iRBC culture well synchronised and enriched in late stages.

The stage, synchrony and growth conditions are determined by the scientific question the experimenter is asking, not by the assay. For this reason, we provide the number of infected erythrocytes and nuclei used in our studies so that other experimenters can aim for similar numbers regardless of the stage and synchrony. For this study we used asexual blood-stages at 36 ± 4 hp.i. We have clarified this in step 11.

- With regards to reproducibility, all experiments were done in replicate (3 Rs) and the statistics appear adequate.

Minor comments:

- Abstract. A closing bracket is missing.

Corrected

- Step 11: Split each sample into 1mL aliquots at ?

Corrected

- The affinity of proteins A/G to IgG antibodies varies based on host species and IgG subtype (see link). This link does not seem to work

Corrected

Revision Plan

- Low bind tubes are mentioned several times. Please clarify whether it refers to low bind protein or low bind DNA. Step 77.

The vendor and catalog number for the low-bind tubes are specified in the Reagents, Materials & Equipment list.

Which was the desired sequencing depth per library? It could be mentioned here. It is mentioned later in "Quantification and statistical analysis" that the initial desired depth was of 40M read pairs, but what was the real depth obtained? from the Figure 4 seems to be less than 17M read pairs per sample.

Thank you for catching that error. The target was 10M read pairs per library but since CUT&RUN is so specific the isotype controls release less DNA and the resulting libraries produces fewer clusters than aimed for leading to slight over sequencing of the remaining samples.

- Step 79. Please clarify/justify why 50 bp paired-end reads were chosen as sequence length.

After excluding the telomere repeats 100 bp (50+50) are sufficient for uniquely mapping 98.3% of the nuclear. Paired-end sequencing was chosen over single-end because it provides the actual size of each fragment.

In the section "Quantification and statistical analysis", references to Figure 3 and 4 are inverted or do not correspond with the actual figures 3 and 4.

Corrected

- Figure 2. Among the replicates, sample 2 seems to have higher background, could you comment why?

It is inherent in biological replicates that one would have the greatest amount of noise but we unfortunately have no further insight into why Sample 2 had a higher elevated background than Samples 1 and 3. Furthermore, even with this slightly higher background the relative enrichment of signal to noise ratio enrichment peaks are readily identifiable.

- Below some suggestions that may help the authors improve the presentation of their data and conclusions:

The limitations and potential shortcomings of the protocol are mentioned along the text (e.g. the use of different antibodies, different targets, weak interactions..), but could be good if they are included in a different section, preferably at the end.

A "Limitations" section was added.

Also in this section it would be good if they develop further (or at least speculate) on the differences in the protocol or things to consider if other type of proteins are assayed (i.e. TFs).

As mentioned above, we have not applied to CUT&RUN to profile chromatin other than Histone PTMs, as this was not the aim of our study. Since chromatin-bound histones always occur within a nucleosomal context, we are hesitant to make claims to the utility of this specific protocol for profiling DNA-binding

Revision Plan

proteins with smaller DNA-binding footprints. That said, CUT&RUN has been used to great success in other systems to profile a wide range of chromatin-bound proteins. We have included mention of this at the end of the introduction.

Authors should better comment on the potential impact of chromatin structure and DNA sequence (i.e. AT richness) on the biased representation of heterochromatic regions in the data, the level of background and the peak calling analysis.

For the enrichment scores, sequence and accessibility biases cancel out since they are the same for both the PTM-specific antibody and the isotype controls.

The coverage of critical loci, like those belonging to clonally variant gene families, should be calculated and examples of tracks included as supplemental figures.

The locus in Figures 2-4 was specifically chose because it contained clonally variant gene families (var & rifin). However, which loci are considered critical largely depends on the experimental question. To allow researchers to investigate their loci of interest we have provided genome-wide enrichment tracks in BigWig format that are available at NCBI Gene Expression Omnibus under accession number GSE210062 as indicated in the "Quantification & Statistical Analysis" and "Data Availability" sections. These tracks can be directly loaded into the PlasmoDB Genome Browser Tool and combined with the many other tracks available there.

Authors claim that the CUT&RUN protocol has exceptionally low background and has been successfully used to profile chromatin interactions from very small numbers of cells. But it is not specified how many. That is, which is the standard in other fields and how it compares with the number of cells used here. As stated in the note following step 10, we did not optimize the minimum number of parasites required in this study since at the $1e7$ iRBC required for each sample correspond as little as 1mL of bloodstage culture 2% parasitemia and 5% hematocrit. The down-sampling analysis in figure 3 suggests that the number of input cells can likely be reduced at least 10-fold.

Information about synchronisation, estimation of iRBCs density and nuclear content appears insufficiently described and has been fragmented in different sections so it is difficult to replicate. For example, within the section "Binding cells to Concanavalin A-coated beads" different alternative protocols for iRBCs synchronisation and enrichment are mentioned but it is not clear whether authors actually perform that step. It could be convenient to describe it and include it in the step-by-step protocol. The stage, synchrony and growth conditions are determined by the scientific question the experimenter is asking, not by the assay. For this reason, we provide the number of infected erythrocytes and nuclei used in our studies so that other experimenters can aim for similar numbers regardless of the stage and synchrony. For this study we used asexual blood-stages at 36 ± 4 hp.i. We have clarified this in step 11.

Significance (Required)

The CUT&RUN is a novel technique to profile chromatin modifications genome-wide that has been successfully adapted to *P. falciparum* by the authors. This technique overcomes important limitations of

Revision Plan

the traditional ChIP-seq and provides better quality data. First, fewer cells and lower sequencing depths are required which is fundamental for the analysis of certain parasite life stages. Second, the binding step is carried out in-situ using unfixed and intact cells. This allows to avoid crosslinking, which can interfere with target recognition that results in unspecific background, and also avoids the random fragmentation of the chromatin, that can bias in the analysis.

This work is significant since it represents the first CUT&RUN step by step protocol adapted to *P. falciparum*. The results are important for researchers from the malaria field and parasitologists in general who could eventually leverage this protocol to other Apicomplexa.

Our expertise is on transcriptional regulation, molecular parasitology, genomics and epigenomics, of malaria parasites. We hope the comments above will help the authors to improve the ms. Congratulations on the work.

Reviewer #3 (Evidence, reproducibility and clarity (Required)):

In general the paper is very clear and convincing and I have only minor comments for the authors to address.

Introduction: The authors state that 'crosslinking presents another challenge as it can interfere with antibody recognition.' Would it be possible to provide a reference to strengthen this statement?

Additional references (Baranello et al, O'Neill et al) were added.

In the third paragraph the authors mention that CUT&RUN can be used to profile chromatin interactions from very small numbers of cells. This argument would be strengthened by adding references or examples from mammalian systems, and the authors might mention that the slight modification CUT&TAG has been employed for single cell sequencing. <https://doi.org/10.1038/s41587-021-00865-z>.

The reference was added as suggested.

Figure 2: A label of Relative fold enrichment should be added to the y axis. This applies also to Figure 4. In the legend, it isn't entirely clear from what control the fold enrichment is being generated. Based on the other figures I assume it's the isotype control, and it would be helpful to state that in the legend.

Thank you for the suggestion. We have made these changes.

Figure 3: An HP1 ChIP is included but there is no track for HP1 using CUT&RUN. It isn't entirely clear to me why HP1 is included; is it to make the point that it overlaps with H3K9me3? There is a sentence at the end of the Quantification and statistical analysis section that indicates that an HP1 CUT&RUN experiment was performed ('Representative tracks of H3K9me3, H3K4me3, and HP1 produced using CUT&RUN or ChIPseq at chromosome 8 of *P. falciparum* are shown in Figure 4), but I don't see an HP1 track for Figure 4 and I don't see CUT&RUN HP1 tracks on Figure 3.

Correct, no HP1 CUT&RUN was performed, we are just trying to show that H3K9me3 CUT&RUN recapitulates ChIP-seq of both H3K9me3 and HP1, which binds to H3K9me3.

Revision Plan

DNA purification by Phenol/Chloroform extraction

In step 38, I noticed that RNase was not added at this step, as described in the original paper by Skene et al. Can the authors make a brief note about why they omit this reagent?

RNase A is already present since it is included in the STOP buffer at step 35.

The authors mark the TE buffer in bold, but I don't see a description of its makeup in the buffer section, though possibly I missed it. While this is a pretty standard buffer, it might still be nice to include it for completeness.

TE buffer recipe was added.

Clean-up of PCR amplified library

Between step 70 and 71 the authors include a warning to not discard the beads. However, this warning is not included in the Post-ligation Clean-up, which involves much the same procedure.

Corrected

Typos and writing

It might be helpful to define CUT&RUN in the abstract by spelling out the acronym there.

It is defined in the 3rd paragraph of the introduction

I've mostly seen ChIP-seq with a dash between the IP and the seq.

Corrected

Powerful is used twice in consecutive sentences in the first paragraph of the introduction. Consider substituting the words 'important tool' for 'powerful tool.'

Corrected

Figure 2 legend: “(purple) in of three biological replicates” should be “(purple) in three biological replicates”

Corrected

Figure 3 legend: Last sentence should include 'to' between the words 'shown' and 'the'.

Corrected

Post-ligation Clean-up: “Wash twice with 200µl of 80% Ethanol freshly prepared” should be “Wash twice with 200µl of freshly prepared 80% Ethanol”

Corrected

Library PCR amplification: “Fragments are PCR amplified using Kapa polymerase, which it is more efficient” should be “Fragments are PCR amplified using Kapa polymerase, which is more efficient”

Corrected

Figure 7 legend: “Indicated in the tope left panel” Should be “Indicated in the top left panel”

Corrected

Revision Plan

Expected outcomes: “to dismiss any sort of contamination” should be “To dismiss contamination”

Corrected

Potential Solution: After the sentence 'Incubation buffer is added.' The next letter 'i' should be capitalized in the word Isolate.

Corrected

CROSS-CONSULTATION COMMENTS

Plasmodium is not my model organism, so I'd defer to Reviewer 2 on the comments regarding additional detail for synchronization and Plasmodium culture conditions. I have nothing further to add, and I'm excited to see what experiments come from the addition of this technique to the parasite field.

Reviewer #3 (Significance (Required)):

This excellent methods paper describes a detailed protocol for the adaptation of the Cleavage Under Targets & Release Using Nuclease (CUT&RUN) technique to Plasmodium falciparum, the causative agent of malaria. CUT&RUN is an alternative to ChIP-seq, and has the advantage that it does not require crosslinking of targets, which can introduce artifacts and cause issues with antibody recognition. CUT&RUN can also be performed with low numbers of cells and has an excellent signal to noise ratio, which the authors demonstrate by downsampling the number of reads used in their analysis. The authors also clearly demonstrate that profiling of histone modifications using CUT&RUN yields comparable results to ChIP-seq. Because it can be difficult to obtain large numbers of cells from Plasmodium cultures, CUT&RUN is especially useful in this important model system. Publication of a detailed protocol will help other Plasmodium researchers answer important questions regarding genomic localization for their targets of interest.

4. Description of analyses that authors prefer not to carry out

The following three suggestions by the reviewers were not incorporated for the reasons indicated.

Reviewer 2: For *P. falciparum* WGS a PCR-free library preparation is strongly recommended. We wonder if it would be possible to try to integrate this step in their CUT&RUN protocol.

Since such biases are sequence dependent, they would impact raw coverage but cancel out in the enrichment plots since the sequence-based biases are identical in both samples. While PCR-free amplification may be desirable for some applications, we feel this both unnecessary for the described application and outside the scope of this study to implement PCR-free amplification.

Reviewer 2: It would have been desirable to have tried the CUT&RUN protocol on other type of proteins, different to hPTMs which are highly abundant, for example one of the Pf Api-AP2 transcription factors. Assaying the CUT&RUN protocol on a different type of protein shouldn't be cost/time consuming and would provide evidence of the versatility of the approach.

As indicated by the title, this protocol was optimized specifically for profiling of histone modifications. CUT&RUN has been used in other systems to profile genome-wide binding of other proteins but this was not our aim and outside the scope of this study.

Reviewer 3: Figure 4: The downsampling of reads is a nice demonstration that low numbers of reads are required for the CUT&RUN technique. It might be helpful to include downsampling of ChIP-seq reads within this figure to compare the two techniques more directly.

The reason we included the down-sampling of CUT&RUN sequence reads was to explore whether were over-sequencing our CUT&RUN libraries not to provide a comparison to ChIP-seq. For simplicity we have therefore kept the figure as is.

October 27, 2022

RE: Life Science Alliance Manuscript #LSA-2022-01778-T

Bjorn F.C. Kafsack
Weill Cornell Medical College
Microbiology & Immunology
1300 York Ave
W-705
New York, NY 10065

Dear Dr. Kafsack,

Thank you for submitting your revised manuscript entitled "Genome-wide Profiling of Histone Modifications in Plasmodium falciparum using CUT&RUN.". We would be happy to publish your paper in Life Science Alliance pending final revisions necessary to meet our formatting guidelines.

- please upload your main manuscript text as an editable doc file
- please upload both your main and supplementary figures as single files
- please add a Running Title, Category, and an alternate abstract to our system
- please add the Twitter handle of your host institute/organization as well as your own or/and one of the authors in our system
- please consult our manuscript preparation guidelines <https://www.life-science-alliance.org/manuscript-prep> and make sure your manuscript sections are in the correct order
- please use the [10 author names, et al.] format in your references (i.e. limit the author names to the first 10)
- please double-check your figure callouts and add a callout for Figure 5 and Figure 6

A. FINAL FILES:

B. MANUSCRIPT ORGANIZATION AND FORMATTING:

Sincerely,

Reviewer #1 (Comments to the Authors (Required)):

This paper provides a detailed step by step protocol of the novel CUT&RUN technique adapted to the malaria parasite *Plasmodium falciparum*. The authors apply the CUT&RUN protocol to infected red blood cells and obtain very good quality genome-wide profiles of two histone modifications, H3K4me3 and H3K9me3. The results were compared with previous ChIP-seq data and show a substantial improvement in terms of lower cell number and sequencing depth required and better signal to noise ratio. The protocol is fully detailed and the figures reflect the quality of the results.

In this revised version of the manuscript the major and minor revisions are fully addressed. The new supplementary figures improve the paper and strengthen the conclusions. No additional changes or experiments are required.

October 28, 2022

RE: Life Science Alliance Manuscript #LSA-2022-01778-TR

Dr. Bjorn F.C. Kafsack
Weill Cornell Medicine
Microbiology & Immunology
1300 York Ave
W-705
New York, NY 10065

Dear Dr. Kafsack,

Thank you for submitting your Methods entitled "Genome-wide Profiling of Histone Modifications in Plasmodium falciparum using CUT&RUN.". It is a pleasure to let you know that your manuscript is now accepted for publication in Life Science Alliance. Congratulations on this interesting work.

DISTRIBUTION OF MATERIALS:

Again, congratulations on a very nice paper. I hope you found the review process to be constructive and are pleased with how the manuscript was handled editorially. We look forward to future exciting submissions from your lab.

Sincerely,
